evolution, genomics, molecular biology

microRNAs, miRNAs, cnidaria, conservation, turnover, microRNA targets

**Authors for correspondence:**
Yehu Moran
e-mail: yehu.moran@mail.huji.ac.il
Ulrich Technau
e-mail: ulrich.technau@univie.ac.at

# Conservation and turnover of miRNAs and their highly complementary targets in early branching animals

Daniela Praher[1], Bob Zimmermann[1], Rohit Dnyansagar[1], David J. Miller[2,3], Aurelie Moya[2,3], Vengamanaidu Modepalli[4,5], Arie Fridrich[4], Daniel Sher[6], Lene Friis-Møller[7], Per Sundberg[8], Sylvain Fôret[9], Regan Ashby[10], Yehu Moran[4] and Ulrich Technau[1]

[1]Department of Neurosciences and Developmental Biology; Faculty of Life Sciences, University of Vienna, Vienna, Austria
[2]Department of Molecular and Cell Biology, Comparative Genomics Centre, James Cook University, Townsville, Queensland, Australia
[3]ARC Centre of Excellence for Coral Reef Studies, James Cook University, Townsville, Queensland, Australia
[4]Department of Ecology, Evolution and Behavior; Alexander Silberman Institute of Life Sciences, The Hebrew University of Jerusalem, Jerusalem, Israel
[5]The Marine Biological Association of the United Kingdom, Citadel Hill, Plymouth, UK
[6]Department of Marine Biology, Leon H. Charney School of Marine Sciences, University of Haifa, Haifa, Israel
[7]Danish Shellfish Centre, DTU Aqua, Technical University of Denmark, Lyngby, Denmark
[8]Department of Zoology, University of Gothenburg, Gothenburg, Sweden
[9]Health Research Institute, Faculty of Education, Science, Technology and Mathematics, University of Canberra, Canberra, Australia
[10]Division of Ecology and Evolution, Research School of Biology, Australian National University, Canberra, Australia

VM, 0000-0003-3099-4969; YM, 0000-0001-9928-9294

MicroRNAs (miRNAs) are crucial post-transcriptional regulators that have been extensively studied in Bilateria, a group comprising the majority of extant animals, where more than 30 conserved miRNA families have been identified. By contrast, bilaterian miRNA targets are largely not conserved. Cnidaria is the sister group to Bilateria and thus provides a unique opportunity for comparative studies. Strikingly, like their plant counterparts, cnidarian miRNAs have been shown to predominantly have highly complementary targets leading to transcript cleavage by Argonaute proteins. Here, we assess the conservation of miRNAs and their targets by small RNA sequencing followed by miRNA target prediction in eight species of Anthozoa (sea anemones and corals), the earliest-branching cnidarian class. We uncover dozens of novel miRNAs but only a few conserved ones. Further, given their high complementarity, we were able to computationally identify miRNA targets in each species. Besides evidence for conservation of specific miRNA target sites, which are maintained between sea anemones and stony corals across 500 Myr of evolution, we also find indications for convergent evolution of target regulation by different miRNAs. Our data indicate that cnidarians have only few conserved miRNAs and corresponding targets, despite their high complementarity, suggesting a high evolutionary turnover.

## 1. Introduction

MicroRNAs are a class of small RNAs involved in regulating a plethora of biological processes including embryogenesis, developmental timing and apoptosis in animals and plants by binding complementary mRNA targets and recruiting protein components leading to translational repression or

target cleavage [1–4]. The roles and functions of miRNAs have been extensively explored in bilaterian organisms such as nematodes, insects and vertebrates.

The phylum Cnidaria (corals, sea anemones, jellyfish and hydras) is the sister group of Bilateria [5–7], and thus provides a unique opportunity to explore the evolution of genetic regulatory mechanisms in the animal kingdom. To date, miRNAs have been investigated only in five cnidarian species (*Nematostella vectensis*, *Exaiptasia pallida*, *Stylophora pistillata*, *Acropora digitifera* and *Hydra magnipapillata*) [8–12] revealing only two miRNAs shared between the anthozoan *Nematostella* and the hydrozoan *Hydra*, which are separated by 650–800 Myr [13]. Notably, there is only one miRNA, miR-100, shared between cnidarians (but absent in *Hydra*) and bilaterians [10,14,15], while more than 30 miRNA families are shared in all bilaterian animals [10]. By contrast, bilaterian miRNA targets are largely not conserved.

The conservation of miRNAs and their mRNA targets in Cnidaria, however, has never been addressed systematically. To this end, the miRNA complements of five additional cnidarian species (members of the class Anthozoa belonging to the orders Actiniaria, sea anemones, and Scleractinia, stony corals) were sequenced and the corresponding miRNA targets predicted.

## 2. Results and discussion

### (a) Sequencing and prediction of miRNAs

We previously sequenced miRNAs of eight developmental stages of the sea anemone *N. vectensis* and identified 87 miRNAs with a custom pipeline [10]. In a recent study combining immunoprecipitation of *Nematostella* Argonaute proteins, high throughput sequencing of small RNAs and annotation of miRNAs by miRDeep2 [16] during the sea anemone's development Fridrich *et al.* provided experimental evidence for the existence of 138 miRNAs [17]. Among these, 52 miRNAs were previously identified [10,14] and 86 miRNAs are novel miRNAs [17].

To explore the conservation and turnover of cnidarian miRNAs, miRNA complements were sequenced from five additional anthozoan species: the sea anemones *Edwardsiella carnea*, *Scolanthus callimorphus*, *Metridium senile* and *Anemonia viridis*, and the stony coral *Acropora millepora*. We annotated miRNAs using the state-of-the-art tool for identification and annotation of miRNAs, miRDeep2 [16] and filtered candidate miRNAs against tRNAs and rRNAs (see Material and methods for details). In the case of cnidarians living in symbiosis with dinoflagellates, *Symbiodinium* miRNAs were filtered out using the available genome data (*S. microadriaticum*, *S. kawagutii* and *S. minutum*) [18–20]. Following the core algorithm of miRDeep2, we additionally ran the package's quantifier module to detect conserved miRNAs not found by the previous strategy and to compensate for incomplete genome assemblies. miRDeep2 initially predicted 56 candidates in *A. viridis* from which 16 are bona fide miRNAs. In *A. millepora* 285 candidates were predicted, which were subsequently reduced to 50 bona fide miRNAs. *A. millepora* is a close relative to *A. digitifera*, for which 26 miRNAs were estimated recently [11]. To provide methodological consistency, we re-annotated the species' miRNAs based on small RNA sequences of *A. digitifera* larvae, two replicates of an adult single colony and two replicates of pooled colonies [11] yielding 292 candidates, from which 42 passed manual inspection using the same criteria as for the other species. We took a similar approach for the published small RNA datasets of the sea anemone *E. pallida* [12] and the stony coral *S. pistillata* [9] miRNA datasets, which yielded 47 and 23 bona fide miRNAs, respectively. For *E. carnea*, *S. callimorphus* and *M. senile* no genome assemblies are currently published. Hence, in these cases, conserved miRNAs were detected using the quantifier module via homology alone. This approach yielded 10 candidates and 7 bona fide miRNAs for *E. carnea*, 17 candidates and 9 bona fide miRNAs for *S. callimorphus* and 13 candidates and 10 bona fide miRNAs for *M. senile*. A comprehensive list of bona fide miRNAs of all species and those who did not pass the criteria can be found in electronic supplementary material, tables S1 and S2 (for species with and without available genome).

Our expanded miRNA repertoire in Anthozoa enabled a reconstruction of the accumulation history of miRNAs in this class. Within anthozoans, Actiniaria and Scleractinia share 13 and 10 miRNAs, respectively (figure 1*a*). The low number ($n = 9$) of miRNAs shared across most anthozoans investigated here is consistent with the previous report of only two conserved miRNAs [8,10] (miR-2022 and miR-2030) within the whole phylum Cnidaria, and reflects remarkably rapid gains and losses of miRNAs. The overwhelming majority of cnidarian miRNAs belong to families of singleton members, the only exceptions being three miRNAs of the 2024 family in *Nematostella* [10] and the duplications of miR-2022 specific to *Exaiptasia* [12]. By contrast to this high turnover of miRNAs in Cnidaria, Bilateria share more than 30 miRNA families. Notably, among the eight conserved anthozoan miRNAs each is shared between at least one scleractinian coral and one actiniarian sea anemone (figure 1*b*), demonstrating their ancient origin.

Until recently, it was assumed that miRNA pathways evolved independently in animals and plants (reviewed in [22]) due to apparent differences in the biogenesis of miRNAs and their target regulation. However, recent comparative analyses of the proteins required for miRNA biogenesis and mechanisms of target regulation call this assumption into question, and suggest the possibility of a common origin of the animal and plant miRNA pathways (reviewed in [15]). In plants, miRNAs are frequently gained and lost throughout evolution, thus few miRNAs are conserved between distantly related plant lineages [23,24] (reviewed in [25]). Comparison of the miRNA turnover rates in plants and Bilateria suggests that the miRNA flux of gains and losses in plants is higher (reviewed in [15]). Comparison of the miRNA repertoires of the closely related *Arabidopsis* species *A. thaliana* and *A. lyrata* implied that between 1.2 and 3.3 miRNA genes are gained or lost per million years [24]. By contrast, the miRNA flux for drosophilids was calculated to be in the range of 0.8–1.6 miRNA genes per million years [26,27]. It is possible to calculate miRNA turnover rates based on the divergence times between species for which miRNA data are available. Due to the absence of a skeleton sea anemones are poorly represented in the fossil record, hence an accurate estimation of divergence times or miRNA turnover rates for these species is impossible. In the case of stony corals, however, the presence of a calcified skeleton provides a robust basis for more reliable estimates of divergence times, which can be used to make inferences about miRNA turnover rates. Based on the assumption that *A. digitifera* and

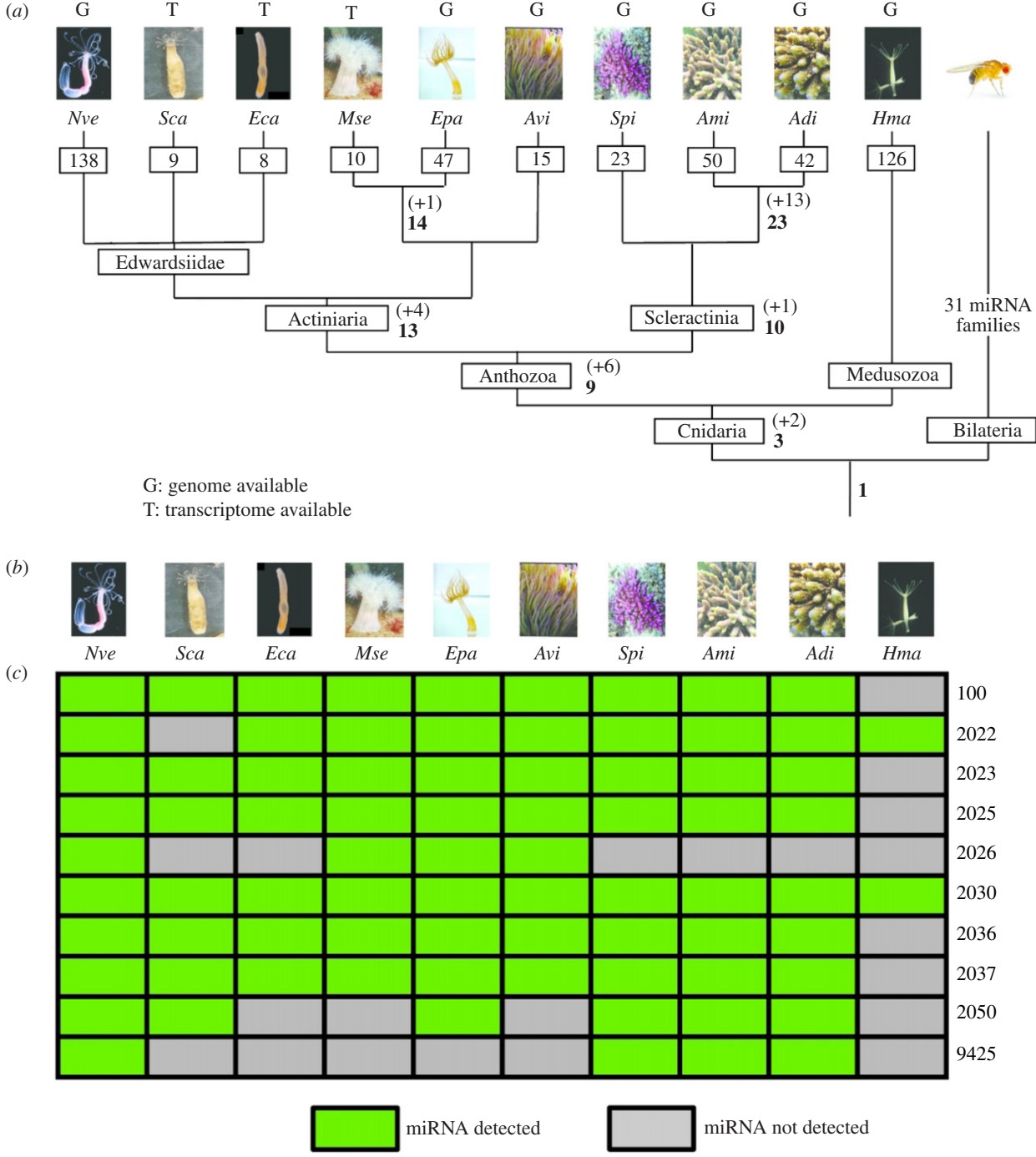

**Figure 1.** Anthozoan miRNAs. (*a*) Evolutionary gains of miRNAs within Cnidaria. Numbers in bold at nodes of the phylogenetic tree represent the total number of shared miRNAs, while numbers in brackets with a+sign indicate gains. Numbers in squares on top of the branches describe total numbers of identified miRNAs in each species. Letter above each species indicates the availability of a published genome ('G' for *Nve, Epa, Avi, Spi, Ami, Adi* and *Hma*) or a transcriptome ('T' for *Sca, Ecal, Mse*) for the respective species. Data from *Hma* were taken from [8]. Abbreviations: *Nve, Nematostella vectensis, Eca, Edwardsiella carnea; Epa, Exaiptasia pallida, Sca, Scolanthus callimorphus; Mse, Metridium senile; Avi, Anemonia viridis; Spi, Stylophora pistillata; Ami, Acropora millepora; Adi, Acropora digitifera; Hma, Hydra magnipapillata; Dre, Danio rerio.* (*b*) Conservation of the core set of anthozoan miRNAs shared between at least four species. Green squares indicate presence of miRNAs whereas grey squares reflect that the miRNA is not expressed [21].

*A. millepora* separated 16.2–35.3 Ma [28], comparison of the presented miRNA repertoires imply that miRNA losses and gains occurred with a rate of 0.7–1.4 genes per million years. While it is tempting to compare the miRNA turnover rates between these different groups of organisms, it is important to note that differences in miRNA annotation criteria and methods between different studies and the blurry distinction of some small interfering RNAs (siRNAs) from miRNAs in insects, cnidarians and plants, would make this comparison inaccurate. Nonetheless, the surprisingly small number of

miRNAs conserved between sea anemones, even within the same family, is consistent with a significant miRNA turnover rate.

Analysis of the nucleotide composition of all anthozoan miRNA sequences identified here reveals a strong bias for U at the first position of the mature sequence, representing a characteristic feature of miRNAs explained by the preference of Argonaute proteins for a U or an A at the 5′ end of the miRNA [29] (figure 2*a*). Interestingly, miR-100 remains the only miRNA shared between Cnidaria and Bilateria

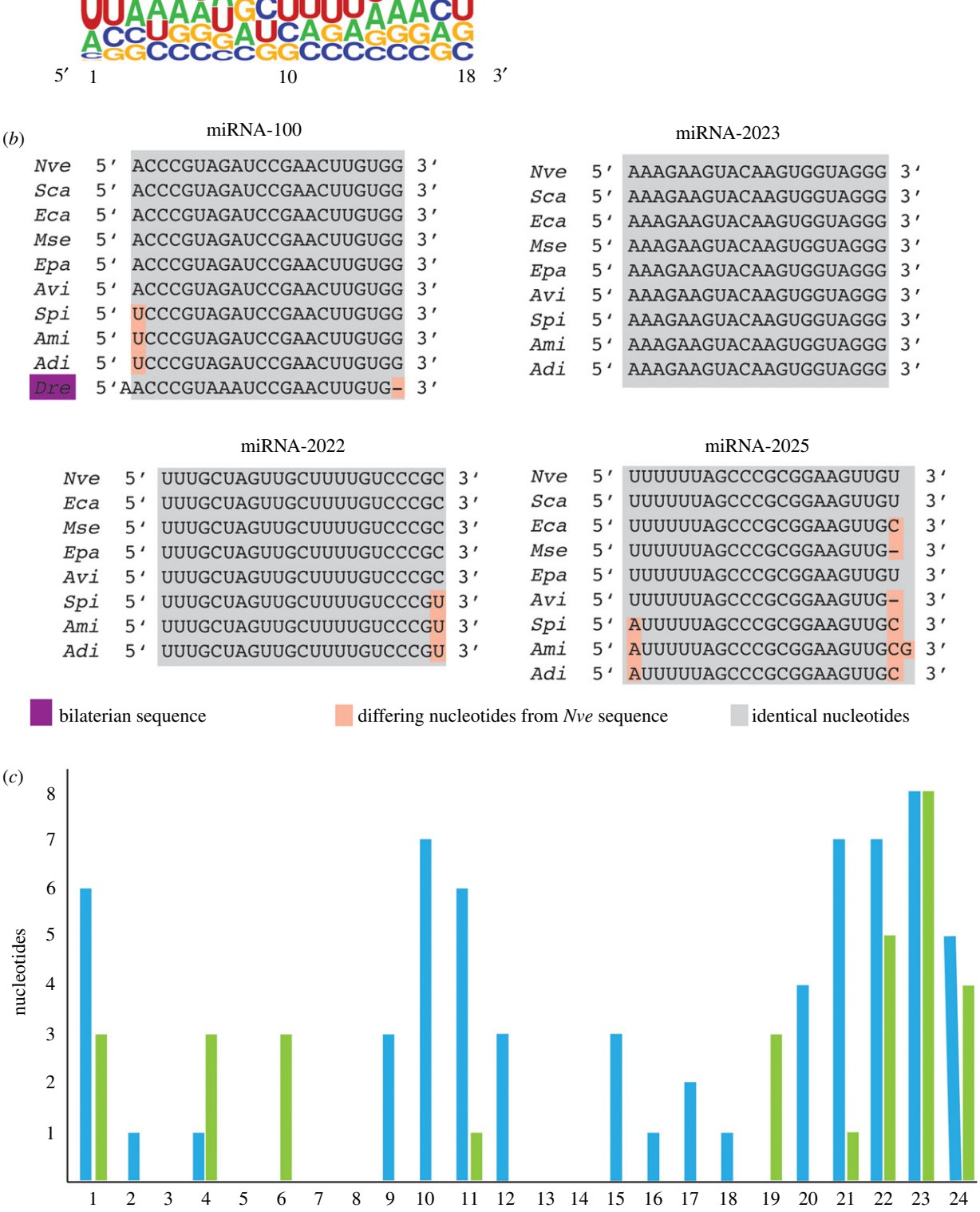

**Figure 2.** Features of anthozoan miRNA sequences. (*a*) Sequence logo of nucleotides bias of anthozoan miRNAs. Despite overall low bias of nucleotides along the miRNA sequences, a notable U bias at position 1 can be detected, which is a known characteristic of miRNAs in bilaterian species. (*b*) Conservation of the mature miRNA sequence exemplified by miRNA-100, miRNA-2022, miRNA-2023 and miRNA-2025 within investigated cnidarian species. Note the low number of nucleotide changes, especially for miRNA-2025 whose sequence does not experience any substitution. (*c*) Comparison of the evolutionary dynamics of miRNA sequences in Bilateria (blue) and Cnidaria (green). For each phylum, nine conserved miRNA genes were analysed. In Bilateria, the sequences gained 62 changes in total, whereas cnidarian sequences only had 31.

and all cnidarian homologues show the previously described 1 nucleotide shift compared to the bilaterian sequence (figure 2*b*) [14].

Cnidarian miRNAs display remarkably high sequence conservation over their whole length. The miR-2023 homologues in Cnidaria show identical sequences to each other

without a single nucleotide substitution. Notably, the few substitutions which occur in other miRNAs are more frequent at the ends rather than in central parts (figure 2*b*). This notion is further supported, when considering the nucleotide substitutions for the conserved core set of cnidarian miRNAs (figure 2*c*): The nucleotides at positions 7–18 largely do not change in the studied species (with position 11 showing a single substitution). By contrast, the bilaterian miRNAs show a high variability of nucleotide composition outside the seed region, especially at positions 9–12 and towards the 3′ end of the miRNAs. In total 62 changes were detected for bilaterian miRNAs, while only 31 changes are observed for the investigated cnidarian miRNAs despite the fact that the same number of species and of miRNAs were compared. As miRNAs attain their biological activity by binding to mRNA targets, we next explored the evolutionary dynamics of cnidarian miRNA targets.

## (b) Prediction of miRNA targets and investigation of miRNA target conservation

Targets of miRNAs can be predicted bioinformatically based on sequence complementarity [30–32] with the risk of identifying many false-positives [33,34]. Alternatively, putative targets can be identified by AGO PAR-CLIP which detects mRNAs bound to Ago-miRNA complexes [12,35], but whether all bound mRNAs are indeed functionally modulated, remains unclear. Here, we aimed to identify putative functional miRNA targets and investigate the extent of conservation of miRNA target relationships in cnidarian species. Our previous work implied that the mechanism by which cnidarian miRNAs bind targets differs fundamentally from the one in bilaterians [10]. In bilaterians, the principal determinant of miRNA binding is the seed sequence, a stretch of seven nucleotides covering positions 2–8 at the 5′ end of the miRNA. Base-pairing of the seed alone is sufficient for binding and affecting the target. Subsequent recruitment of protein complexes cause translational inhibition and transcript decay of potentially many target mRNAs [3,36]. Contrastingly, cnidarian miRNAs frequently exhibit extensive complementarity to their targets, leading to miRNA-mediated target cleavage [10]. Thus, cnidarian miRNAs resemble those of plants in terms of their mode of action (reviewed in [15]). While cnidarian miRNAs might also target some mRNAs via shorter matches of 12–15 nucleotides [12] and probably also employ translation inhibition as a mechanism of action [37], such shorter matches would be difficult to detect bioinformatically without risking a high false-positive rate. Hence, we chose to use the extensive complementarity of cnidarian miRNAs to their sliced targets found in degradome and RLM-RACE experiments as criterion [10]. These datasets enabled the use of a custom pipeline rather than conventional prediction tools in order to find miRNA targets shared between *Nematostella* and other species investigated. We took advantage of the verified miRNA target interactions in *Nematostella* and *H. magnipapillata* [10] and extracted stringent targeting rules by visualizing the binding properties of cnidarian miRNAs and their functionally modulated targets by the using RNAhybrid [38]. *Bona fide* miRNAs were mapped to the respective transcriptome by applying these rules using a customized version of an established mapping algorithm [39]. A set of homologous targets was selected based on reciprocal best BLAST hits to *Nematostella* genes whose target

sites were verified using previously published degradome data [10]. Putative targets resulting from this analysis were, like their *Nematostella* and *Hydra* miRNA targets previously, subjected to deeper inspection by RNAhybrid [38] and removed if they did not fulfil the set rules.

This stringent approach enabled the identification of a number of homologous mRNA targets between the species (see electronic supplementary material, table S3 for details of verified shared targets among the investigated species). Homologous targets were considered to be 'conserved' if they are regulated by the same miRNA (category I in figure 3), as distinct from 'shared' targets, which are targeted by different miRNAs between the species (category II in figure 3). Some additional miRNA target pairs that were identified could be real targets *in vivo*, but did not pass our stringent RNAhybrid rules (category III in figure 3).

All conserved miRNA targets keep the position of the miRNA binding site within the mRNA through cnidarian evolution and all of them except for one (*HoxD*) are situated in the coding region of the respective target gene (figure 4*a*). Coding sequences are generally the most conserved regions within a gene and the presence of miRNA binding sites in these might be expected to impede the accumulation of mutations even at third codon positions, leading to conservation during evolution. The sequences of the miRNA binding sites are extremely conserved, not only in the seed sequence, exhibiting only very few (one or two) nucleotide changes over the length of the binding site as exemplified by *Six3/6* and miR-2025 (figure 4*b*). This is consistent with the notion that the seed sequence of cnidarian miRNAs alone is probably not sufficient for target regulation, in contrast to Bilateria [10,12]. The number of nucleotide substitutions increases with evolutionary distance from *N. vectensis*: In *S. callimorphus* and *E. carnea* a single substitution or none can be found in the miRNA binding site respectively, while in *A. viridis* and stony corals two substitutions are detected. The conservation of the genomic location and the sequence identity of miRNA binding sites in targets indicate that these target sites were already present in the common ancestor of the two species. Furthermore, considering the evolutionary distances between the species (e.g. approx. 500 Myr between *Nematostella* and *Acropora*) [40], the conservation of nearly identical miRNA binding sites between *Nematostella* and other anthozoan species strongly implies functionality. This assumption is supported by the fact that all conserved miRNA targets identified here (*Six3/6*, *HoxD*, *nematogalectin-like 2* and *2030T* are cleaved after guidance by the respective miRNA as previously demonstrated in *Nematostella* [10].

The number of conserved miRNA targets ranges between three and seven with *Six3/6* and the *HoxD* being the most conserved ones in all anthozoan species sequenced (figures 3 and 4). Interestingly, no evolutionary trend like the expected decrease of conserved miRNA targets with increasing evolutionary distance could be observed. Our analysis rather indicates that the number of conserved miRNA targets stay similarly low in all species. Despite a deep conservation of many miRNA families in Bilateria, very few examples of miRNA target conservation have been identified unequivocally, and several studies show that target site conservation is barely more significant than accidental occurrence (reviewed in [41]). This is probably connected to the fact that a typical bilaterian miRNA confers target regulation followed by binding its targets via the seed sequence of seven

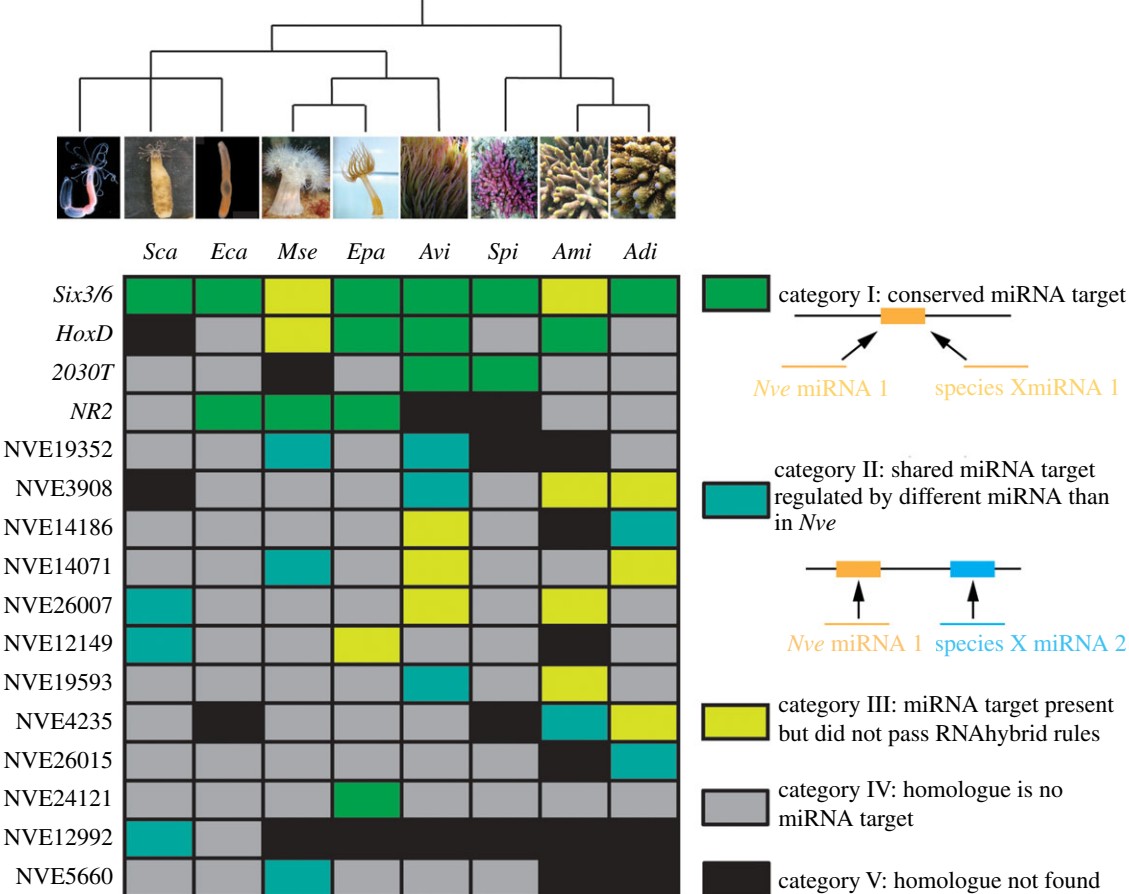

**Figure 3.** miRNA targets in Cnidaria. (*a*) Conserved and shared miRNA targets between *Nve* and at least one other investigated species. miRNA targets are labelled with their gene model annotation of the *Nve* homologues. Blast hits for the gene models are: NVE12346: *Six3/6*, NVE21156: *HoxD*, NVE19315: *uncharacterized protein* (previously described as *2030T* [10], NVE18870: *uncharacterized protein* (previously described as *Nematogalectin related 2* (*NR2*; [10]), NVE19352: *fer1l3 protein*, NVE3908: *serine/threonine-protein kinase PRP4*, NVE14186: *F-actin-capping protein subunit beta*, NVE14071: *f-box wd repeat-containing protein 4*, NVE26007: *centrosomal protein of 290 kDa-like*, NVE12149: *eukaryotic translation initiation factor 3 subunit j*, NVE19593: *E3 ubiquitin-protein ligase IRF2BPL*, NVE 4235: *delta alicitoxin-Pse2b*, NVE26015: *uncharacterized protein*, NVE24121: *cAMP-dependent protein kinase type I-alpha regulatory subunit*, NVE12992: *titin-like*, NVE5660: *heterogeneous nuclear ribonucleoprotein A0*.

nucleotides. Such a miRNA binding site can be found by pure chance and the length of a transcripts' 3′ UTR is directionally proportional with this probability. A well-documented case of conservation of a miRNA target pair is miR-196 targeting *HoxB8* in vertebrates [42]. This perfectly matching miRNA-mRNA pair is conserved between fish and mammals, lineages with an estimated divergence time of approximately 400–450 Myr [43], and hence represents one of the oldest conserved miRNA target relationships in Bilateria. Strikingly, the four miRNA target interactions conserved between sea anemones and stony corals found in this study (figures 3 and 4*a*) most probably exceeds this age, since they date back to the last common anthozoan ancestor, approximately 500–650 Ma [13,40,44].

By contrast to the limited extent of conservation in Bilateria, there are many well-documented cases of miRNA target conservation between distantly related plant phyla [45–48]. For example, 35 miRNA targets are conserved between the bryophyte *Marchantia polymorpha* and the angiosperm *Arabidopsis thaliana*, taxa that diverged approximately 450 Ma [48]. The observed differences in target site conservation between bilaterian animals and plants could be connected to the mechanisms by which miRNAs bind their targets in the two groups. miRNA target sites in plants might be under higher selective pressure because they are longer and are often within protein-coding regions, whereas in bilaterians the seed regions are much shorter and typically bind in the 3′-UTR.

Our investigation implies that miRNA target conservation is probably lower in cnidarians than in plants. This notion provides an interesting twist: As mentioned above, cnidarian miRNAs regulate their targets by a cleavage mechanism enabled by high complementarity between the miRNA and its target. While this mechanism is very similar to the mode of action of plant miRNAs, target site conservation seems to differ dramatically between cnidarians and plants. The reason for this difference remains an open question. One possible explanation is that miRNA target recognition may be more promiscuous in Cnidaria than in plants as indeed suggested by a recent PAR-CLIP analysis of a cnidarian Argonaute [12]. Alternatively, the high conservation of miRNA target sites observed in plants may be a cause rather than a consequence of the miRNA targeting mechanism. The extent of target conservation in cnidarians seems to be comparable to that observed in bilaterians or even higher, as we did not include conserved targets that are not as stringent as those caught by the degradome sequencing in our analysis. Hence, it is possible that the mode of miRNA target binding in Cnidaria and the pace of target evolution in this phylum represent an intermediate between the plant and bilaterian

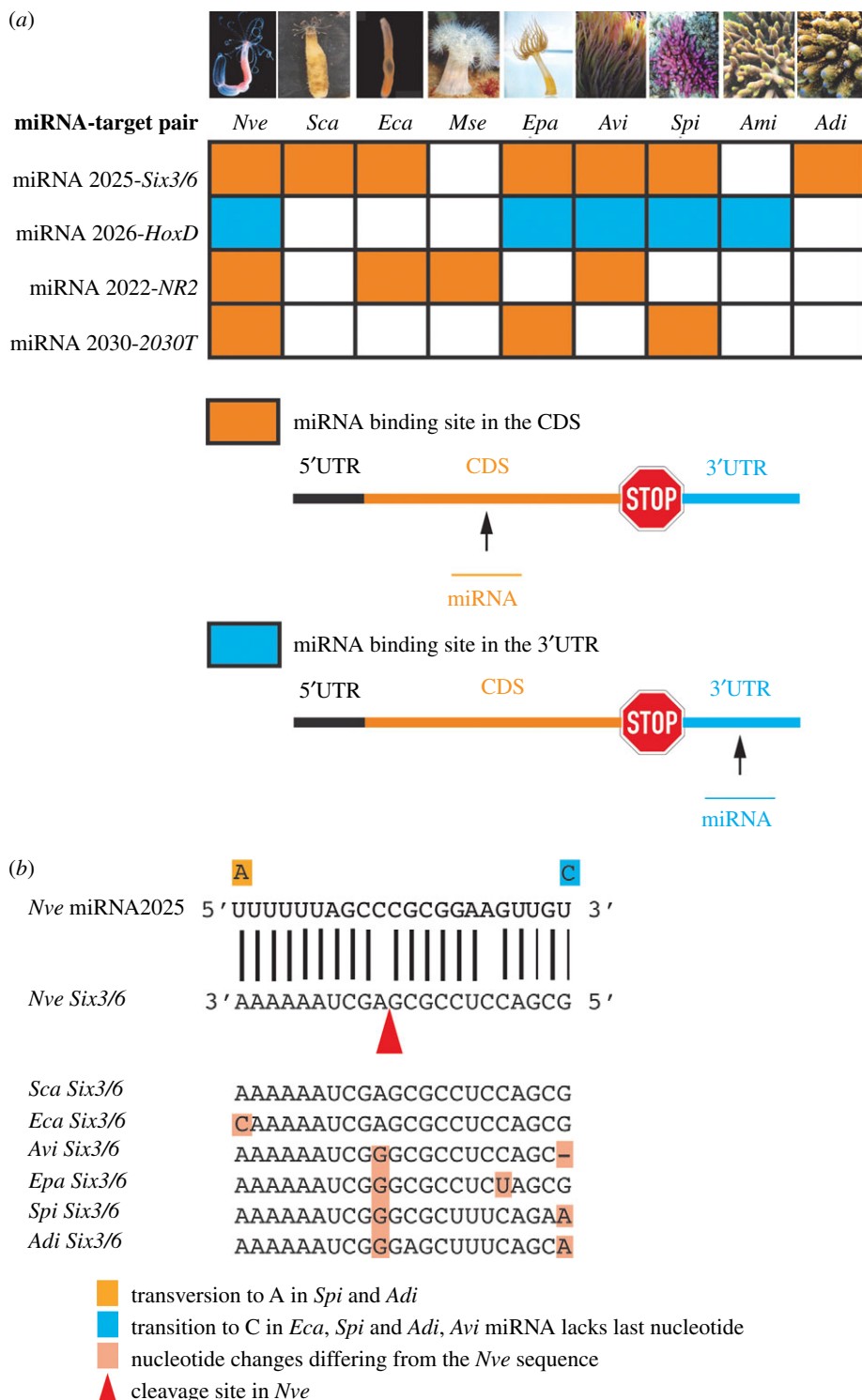

**Figure 4.** Conservation of miRNA binding sites in Cnidaria. (a) The conserved miRNA targets are regulated by the same miRNA in *Nve* and other investigated species and keep their genomic location of the miRNA binding site. Abbreviations: UTR, untranslated region. (b) Conservation of the miRNA binding site in *Six3/6*, target of miRNA-2025, in investigated species. *Six3/6* is cleaved in *Nve* (position indicated by the arrowhead). This represents the most ancient miRNA target interaction known to date. Abbreviations: *Nve, Nematostella vectensis, Eca, Edwardsiella carnea; Epa, Exaiptasia pallida; Sca, Scolanthus callimorphus; Mse, Metridium senile; Avi, Anemonia viridis; Spi, Stylophora pistillata; Ami, Acropora millepora; Adi, Acropora digitifera.*

states. However, testing of this notion will require further investigation of miRNA target binding in Cnidaria.

The analysis also revealed that several cases where shared miRNA target transcripts are bound by different miRNAs between *Nematostella* and the compared species. In all these cases, the miRNA employed for target regulation in *Nematostella* was not detected in the second species, and the binding of the miRNA target was taken over by different miRNAs. This strongly suggests independent evolution of

miRNA-mediated regulation of the same targets. Apparently, these miRNA targets are under selective pressure to be regulated, but which specific miRNA exerts the action is less important. Thus, the putative regulation can be carried out by a different miRNA binding to a different target site within the target message. Different scenarios can explain the independent evolution of miRNA target regulation (electronic supplementary material, figure S4). One possibility is that, for some genes, miRNA regulation evolved after the

species diverged, and that at some point post-divergence, distinct miRNAs and miRNA binding sites co-evolved in the descendent lineages to provide regulation. Alternatively, the common ancestor of the two lineages might have had a miRNA binding site, which was retained only in one lineage. In the sister lineage, this miRNA binding site degenerated by genetic drift after a new binding site for a different miRNA evolved. This scenario involves a transient stage, in which the target mRNA had miRNA binding sites for both miRNAs. These results suggest a highly dynamic nature of the evolution of miRNA targets in Cnidaria.

To conclude, we provide the first strategic investigation of the conservation of miRNAs and their targets in the phylum Cnidaria. By sequencing and annotating miRNAs in six species as well as re-annotation in two others, the miRNA complement in Anthozoa expanded significantly and we were able to identify novel and conserved miRNAs. We show that the cnidarian miRNA repertoire is highly dynamic. Further, putative conserved miRNA target interactions were identified—some that are likely to involve conserved miRNAs binding to orthologous targets, and others that involve different miRNAs binding to orthologous targets. Furthermore, we find the most ancient conserved miRNA target relationships known to date, which are shared between sea anemones and stony corals, two groups that separated at least 500 Ma. Our study reveals a very dynamic miRNA repertoire with few deeply conserved miRNAs, most of which arose in the common ancestor of anthozoans. A recent work showed high turnover rates of miRNAs also in scyphozoans [49]; thus, future work should focus to identify and compare miRNAs targets in medusozoans.

# 3. Material and methods

## (a) Collection of animals

Individuals of *A. viridis* were collected in Sdot Yam, Israel, *A. millepora* in Heron Island, Australia, *M. senile* in Kristineberg, Sweden and *S. callimorphus* in Roscoff, France. The individuals of *E. carnea* were isolated from the ctenophore *Mnemiopsis leidyi* collected at the Gullmar Fjord in Sweden.

## (b) Small RNA library preparation and sequencing

Total RNA of three replicates of larvae, two replicates for polyps and six replicates of adult *A. millepora*, two replicates of *S. callimorphus* adult individuals and two replicates each of six *E. carnea* pooled parasitic individuals was extracted with Trizol reagent (Thermo Fisher Scientific Inc.) according to the manufacturer's protocol. Total RNA extraction of *M. senile* was performed as described elsewhere [50] and total RNA of *A. viridis* was isolated with Tri-Reagent (Sigma-Aldrich Inc.) and purified using an RNA Clean-up kit (Zymo Research Inc.). Small RNAs of *A. millepora* were size selected and the library preparation was carried out with the NEBNext Multiplex Small RNA Library Prep Set (New England BioLabs, NEB) following the manufacturer's instructions. The small RNA library was validated on a Bioanalyzer 2100 (Agilent Inc.). Sequencing was performed at the Biomolecular Research Facility (John Curtin School of Medical Research, Australia) on an Illumina HiSeq 2500 platform. Library preparation and small RNA sequencing for the rest of the species was carried out at Exiqon, Denmark. Briefly, total RNA was converted into small RNA libraries using NEBNext library generation kit (NEW England BioLabs Inc.) according to the manufacturer's instructions. The quality of the libraries was evaluated by a Bioanalyzer 2100 (Agilent, Inc.). The libraries were size fractionated on a LabChip XT (Perkin Elmer Inc.) and bands excised. Sequencing was accomplished on an Illumina HiSeq 2500 platform.

## (c) Annotation of miRNAs

For *N. vectensis* sequences of *bona fide* miRNAs were taken from recent Argonaute immunoprecipitation results (PRJNA658931 [17]). miRNA identification for species with available genomes [40,51–53] was performed using miRDeep2 [16]. The adapter trimmed reads of *A. digitifera* (PRJNA298449) and small RNA reads *S. pistillata* and *E. pallida* were downloaded from the SRA (PRJNA234072 and SRX1351928 respectively). The following procedure was essentially identical for all species. Small RNA reads of the replicates and different stages were pooled. Adapter sequences were trimmed (ATCTCGTATGC for *S. pistillata*, TGGAATTCTCGGGTGCCAAGG for *E. pallida* and *A. millepora* AGATCGGAAGAGCACACGTCT for *E. carnea*, *M. senile* and *S. callimorphus*) and reads processed to discard short reads with scripts included in the miRDeep2 package. The remaining reads were mapped to the respective genome using Bowtie (v.0.12.7 as part of the miRDeep2 package) and a list of miRNAs known from other cnidarian species [8–11,54] as input file. A script shared by Yi Jin Liew was used to filter out candidates with a miRDeep2 score below 10 (B.Z. 2017, personal communication) [9]. These candidates were validated and further filtered manually according to criteria summarized by Fromm [54]. These criteria include the requirement of a 2-nucleotide overhang on the 3′ end of the precursor miRNA, 5′ consistency of the mature miRNA strand (at least 90% of the reads have to be starting from the same position), and at least 16 nucleotide complementarity between mature and star strand. In addition, criteria of significant RNAfold *p*-value and a minimum of 10 reads had to be fulfilled to qualify as *bona fide* miRNA. However, the requirement for the terminal loop size of precursor miRNAs above 8 nucleotides and consistency of the star strand of miRNAs were not considered since cnidarian miRNAs do not seem to follow these rules [9]. From all miRNA candidates predicted by miRDeep2 tRNAs and rRNAs were removed by tRNAscan-SE 1.3.1 [55] and sortmerna 2.1 [56], respectively. Filtering of miRNAs expressed from the cnidarian symbionts *S. kawagutii*, *S. minutum* and *S. microadriaticum* [18–20] was performed by mapping miRNA candidates of analysed species to the symbionts genomes using the mapper script of miRDeep2 based on bowtie (v.0.12.7). To compensate for incomplete genome assemblies, the quantifier module was applied to the raw reads of small RNAs of species with available genome. The criteria for validation of the candidates obtained by this module were essentially the same as described above. Since no genome assemblies are available for *E. carnea*, *S. callimorphus* and *M. senile*, we determined solely conserved miRNAs with a minimum of 10 reads with the quantifier module. An overview of miRNA read tracking in each species and miRDeep2 output files of accepted miRNAs are provided in electronic supplementary material, tables S5 and S6–S10, respectively.

## (d) Counting substitutions in conserved miRNA sequences

The nine miRNAs conserved in anthozoans (miR-100, miR-2022, miR-2023, miR-2025, miR-2026, miR-2030, miR-2036, miR-2037 and miR-9425) were compared to identical numbers of bilaterian species (*Branchiostoma floridae*, *Danio rerio*, *Daphnia pulex*, *Drosophila melanogaster*, *Homo sapiens*, *Lottia gigantea*, *Platynereis dumerilii*, *Saccoglossus kowalevskii* and *Strongylocentrotus purpuratus*) and miRNAs (miR-1, miR-7, miR-8 (aka −141 and −200), miR-29 (−83, −235), miR-31, miR-34, miR-92, miR-100 and miR-

375). miRNAs were aligned and each potential site was counted for substitutions. Sequences were taken from [57].

## (e) Prediction of miRNA targets

Transcriptomes of *E. pallida*, *S. pistillata*, *A. millepora*, *A. digitifera* and *A. viridis* were assembled previously [40,52,58,59]. Total RNA of three pooled *M. senile* individuals, six pooled parasitic individuals of *E. carnea* and one individual of *S. callimorphus* was extracted with Trizol (Thermofisher Scientific). Library preparation and transcriptome sequencing was performed at GENEWIZ on an Illumina HiSeq 2000 platform for *M. senile* and for *S. callimorphus*, and at VBCF on a Hiseq 2500 platform for *E. carnea*. De novo transcriptomes for *E. carnea*, *M. senile* and *S. callimorphus* were assembled with the Trinity package [60]. To assess the quality of the transcriptomes, BUSCO analysis was performed [61] (electronic supplementary material, file S11). For miRNA target prediction, we defined rules for the targeting mode of cnidarian miRNAs according to the previously verified miRNA target interactions in *Nematostella* and *H. magnipapillata* [10]. We applied the rules -c -s 11 -S 2 -m 2 -n 1 to a modified version of the sRNA mapper algorithm [39] (mapping bona fide miRNAs to the transcriptome of the respective species. Option -c determines that the 5′ nucleotide of the miRNA does not need to match the target; -s 11 sets the miRNA seed length to 11 nucleotides; -S 2 allows 2 mismatches between the miRNA and its putative target in the miRNA seed sequence, -m 2 allows 2 mismatches between the miRNA and the target in the sequence downstream of the seed sequence and -n 1 allows the last nucleotide of the miRNA to be unpaired to the target sequence.

We determined verified miRNA targets pairs in *Nematostella* using published degradome data [10]. Reads were trimmed using cutadapt with the options -m 10 -q 10,15 –trim-n, with the adapter sequence TCTACAGTCCGA. Reads were mapped to the *Nematostella* genome using bwa [62]. Annotated mRNAs containing a putative miRNA target site which was exactly 10 bp downstream and antisense of a degraded read's 5′ end mapping location were considered as verified targets. Homologous miRNA targets were identified by running reciprocal BLAST searches against a set of the verified target sites of *Nematostella*. We validated the obtained homologous miRNA targets by assessing the annealing properties to their miRNA with RNAhybrid [38] and only miRNA target pairs passed which obeyed the defined rules.

Data accessibility. The transcriptomes of *E. carnea*, *S. callimorphus* and *M. senile* were submitted to the sequence read archive (SRA) under accession PRJNA430035 and the miRNA samples for *A. millepora*, *E. carnea*, *A. viridis*, *S. callimorphus* and *M. senile* are available under accession PRJNA430416.

Authors' contributions. D.P. generated and analysed the data from sea anemones and wrote the draft for the manuscript; B.Z. analysed the data; R.D. assisted in data analysis; D.J.M., A.M., S.F. and R.A. generated the *A. millepora* genomic and small RNA data; L.F.-M. and P.S. collected *E. carnea* samples; V.M. and A.F. assisted with miRNA annotations; D.S. collected *A. viridis* samples and provided genomic data; Y.M. and U.T. designed and supervised the study, and edited and revised the manuscript.

Competing interests. We declare we have no competing interests.

Funding. This work was supported by grants of the Austrian Science Fund FWF (grant nos. P22618, P24858) to U.T., European Research Council Starting Grant (CNIDARIAMICRORNA, 637456) to Y.M., a PhD completion fellowship of the University of Vienna to D.P. and a grant of the Australian Research Council (grant no. CE140100020) to D.J.M. The collection of *S. callimorphus* was supported by Assemble grant 227799 to U.T.

Acknowledgments. We thank the VBCF sequencing facility for transcriptome sequencing.

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
