## [Peer Review File · Proceedings of the Royal Society B: Biological Sciences]

Review History

RSPB-2019-2564.R0 (Original submission)

Review form: Reviewer 1

Recommendation

Reject – article is scientifically unsound

Scientific importance: Is the manuscript an original and important contribution to its field?

Good

General interest: Is the paper of sufficient general interest?

Good

Quality of the paper: Is the overall quality of the paper suitable?

Good

Is the length of the paper justified?

Yes

Should the paper be seen by a specialist statistical reviewer?

No

Do you have any concerns about statistical analyses in this paper? If so, please specify them explicitly in your report.

Yes

It is a condition of publication that authors make their supporting data, code and materials available - either as supplementary material or hosted in an external repository. Please rate, if applicable, the supporting data on the following criteria.

Is it accessible?

Yes

Is it clear?

Yes

Is it adequate?

Yes

Do you have any ethical concerns with this paper?

No

Comments to the Author

Praher and collaborators characterized the evolutionary history of microRNAs in cnidarians and concluded that the turnover of microRNAs and their targets resembles more to that of plant microRNAs than to animals. This is explained in the context that plant and animal microRNAs may have a common evolutionary origin and cnidarians could show features of both. They also sequenced, discovered and annotated new microRNAs in cnidarians, which is always a valuable contribution to the microRNA community.

Although I still believe that the evidence supports a distinct evolutionary origin of microRNAs in plants and animals, I also believe that works like this, challenging the current paradigm, are important and must be always considered for publication. After all, the authors are known and respected and their view and analyses are always welcomed. However, in this occasion I think there is a major issue in the analysis that must be addressed (see below). For that reason, I cannot recommend its publication and I suggest to the editor that he rejects the publication of the paper in its current form. Nevertheless, if the authors consider my concerns and the results still hold, I'll be happy to evaluate this paper again.

MAJOR COMMENTS

The main conclusion that microRNA turnover in cnidarians is similar to that in plants (that is, very high) compared to animals, is based on a strict criteria to determine microRNA orthology. The authors consider only those microRNAs annotated in the miRDeep2 pipeline which can be very strict. On the contrary, other animal microRNAs have been annotated using more 'relaxed' criteria (that is, BLAST) followed from extensive manual curation. That may result in a different annotation depth between this and other studies. Specifically, these are the things I think the authors should consider:

- The coverage of the genome sequence of cnidarians is lower than that of other animals. The authors acknowledge that but, how would this affect to any annotation/comparison? A more in depth comparative analysis, accounting for differences in sequencing coverage, is needed.
- The authors claim (here and elsewhere) that cnidarian microRNAs show features of animal and plant microRNAs. However, the annotation criteria used is that of Fromm et al, which is specifically designed for vertebrates (as a matter of fact, it was originally devised for human microRNAs). How would this affect to the annotation of microRNAs in Cnidarians?

- The authors evaluate cnidarian microRNA targets by long pairwise complementarity, like in plants, but they also acknowledge that short targets (like in animals) can have also an impact. Wouldn't this contradict the main message of the paper?

- Last, but not least, I selected one example of a microRNA that is conserved in some cnidarians but not in others: mir-2050. This microRNA is not detected in *Stylophora pistillata* with the author's methodology. However, a genomic blast reveals that a putative ortholog is detectable in the genome:

```
Spi-mir-2050 -
CCACCAGCUCAAAGCAAUUAAGAUCUUGCUAUGUUUGAUUGCUGUGAUCUGGUUG
UU-
nve-mir-2050
GAACCCGGGUCCCAGUCAUCAAAACGUCAUUGCACAUUUGAUUGCUGUGAUCUGGUUU
UCC
```

This sequence is from a predicted transcript XM_022940493.1, and the RNAfold program shows that it consistently folds into a hairpin:

```
1
GAACCCGGGUCCCAGUCAUCAAAACGUCAUUGCACAUUUGAUUGCUGUGAUCUGGUUU
UCC
1  (((((((((((((((((((((((((((((((((((((((((((((((((((((((((((((((
((((((((((((((((((((((((((((((((((((((((((((((((((((((((((((((((
))))))))))))))))))))))))))))))))))))))))))))))))))))))))))))))
```

Also, the mature sequence UUUGAUUGCUGUGAUCUGGU in Spi is found (in a very rough search) over 1500 times in the expression dataset SRR1130519, which suggest that this microRNA is conserved and expressed in *S. pistillata*. This example suggest that a revised analysis is needed before making strong claims about the conservation pattern of microRNAs in cnidarians.

MINOR COMMENTS

- Reference formatting is inconsistent (sometimes numbering, sometimes Authors' name) and at least two references are missing for the list. Importantly, Moran et al 2017's review is not in the Reference list!

- The description of conserved target (lines 183-186) is not entirely clear to me. The figure associated is not much clearer in this context.

Review form: Reviewer 2 (Kevin Peterson)

Recommendation

Accept with minor revision (please list in comments)

Scientific importance: Is the manuscript an original and important contribution to its field?

Excellent

General interest: Is the paper of sufficient general interest?

Excellent

Quality of the paper: Is the overall quality of the paper suitable?

Good

Is the length of the paper justified?

Yes

Should the paper be seen by a specialist statistical reviewer?

No

Do you have any concerns about statistical analyses in this paper? If so, please specify them explicitly in your report.

No

It is a condition of publication that authors make their supporting data, code and materials available - either as supplementary material or hosted in an external repository. Please rate, if applicable, the supporting data on the following criteria.

Is it accessible?

No

Is it clear?

No

Is it adequate?

No

Do you have any ethical concerns with this paper?

No

Comments to the Author

See attached. (See Appendix A)

Decision letter (RSPB-2019-2564.R0)

02-Dec-2019

Dear Dr Moran:

I am writing to inform you that your manuscript RSPB-2019-2564 entitled "Conservation and turnover of miRNAs and their highly complementary targets in early branching animals" has, in its current form, been rejected for publication in Proceedings B.

This action has been taken on the advice of referees, who have recommended that substantial revisions are necessary. With this in mind we would be happy to consider a resubmission, provided the comments of the referees are fully addressed. However please note that this is not a provisional acceptance.

1) A 'response to referees' document including details of how you have responded to the comments, and the adjustments you have made.

- 2) A clean copy of the manuscript and one with 'tracked changes' indicating your 'response to referees' comments document.
- 3) Line numbers in your main document.

Sincerely,
 Professor Hans Heesterbeek
 mailto: proceedingsb@royalsociety.org

Associate Editor
 Comments to Author:

I enjoyed reading this manuscript by Praher et al. I believe is an important contribution to our understanding of the biology and evolution of miRNAs and cnidarians.

The referees have highlighted methodological issues that are significant, such as the annotation of the miRNAs and data availability. I would like to invite the authors to address them thoroughly before resubmitting the manuscript.

Reviewer(s)' Comments to Author:

Referee: 1

Comments to the Author(s)

Praher and collaborators characterized the evolutionary history of microRNAs in cnidarians and concluded that the turnover of microRNAs and their targets resembles more to that of plant microRNAs than to animals. This is explained in the context that plant and animal microRNAs may have a common evolutionary origin and cnidarians could show features of both. They also sequenced, discovered and annotated new microRNAs in cnidarians, which is always a valuable contribution to the microRNA community.

Although I still believe that the evidence supports a distinct evolutionary origin of microRNAs in plants and animals, I also believe that works like this, challenging the current paradigm, are important and must be always considered for publication. After all, the authors are known and respected and their view and analyses are always welcomed. However, in this occasion I think there is a major issue in the analysis that must be addressed (see below). For that reason, I cannot recommend its publication and I suggest to the editor that he rejects the publication of the paper in its current form. Nevertheless, if the authors consider my concerns and the results still hold, I'll be happy to evaluate this paper again.

MAJOR COMMENTS

The main conclusion that microRNA turnover in cnidarians is similar to that in plants (that is, very high) compared to animals, is based on a strict criteria to determine microRNA orthology. The authors consider only those microRNAs annotated in the miRDeep2 pipeline which can be very strict. On the contrary, other animal microRNAs have been annotated using more 'relaxed' criteria (that is, BLAST) followed from extensive manual curation. That may result in a different annotation depth between this and other studies. Specifically, these are the things I think the authors should consider:

- The coverage of the genome sequence of cnidarians is lower than that of other animals. The authors acknowledge that but, how would this affect to any annotation/comparison? A more in depth comparative analysis, accounting for differences in sequencing coverage, is needed.

- The authors claim (here and elsewhere) that cnidarian microRNAs show features of animal and plant microRNAs. However, the annotation criteria used is that of Fromm et al, which is specifically designed for vertebrates (as a matter of fact, it was originally devised for human microRNAs). How would this affect to the annotation of microRNAs in Cnidarians?

- The authors evaluate cnidarian microRNA targets by long pairwise complementarity, like in plants, but they also acknowledge that short targets (like in animals) can have also an impact. Wouldn't this contradict the main message of the paper?

- Last, but not least, I selected one example of a microRNA that is conserved in some cnidarians but not in others: mir-2050. This microRNA is not detected in *Stylophora pistillata* with the author's methodology. However, a genomic blast reveals that a putative ortholog is detectable in the genome:

```

Spi-mir-2050  -
CCACCAGCUCAAAGCAAUUAAAGAUCUUGCUAUGUUUGAUUGCUGUGAUCUGGUUG
UU-
nve-mir-2050
GAACCCGGGUCCCAGUCAUCAACGUCAUUGCACAUUUGAUUGCUGUGAUCUGGUUU
UCC
          ** ** ** ** ** ** ** ** * ***** *

```

This sequence is form a predicted transcript XM_022940493.1, and the RNAfold program shows that it consistently folds into a hairpin:

```

1
GAACCCGGGUCCCAGUCAUCAACGUCAUUGCACAUUUGAUUGCUGUGAUCUGGUUU
UCC
1  (((((((((((((((((((((((((((((((((((((((((((((((((((((((((((

```

Also, the mature sequence UUUGAUUGCUGUGAUCUGGU in Spi is found (in a very rough search) over 1500 times in the expression dataset SRR1130519, which suggest that this microRNA is conserved and expressed in *S. pistillata*. This example suggest that a revised analysis is needed before making strong claims about the conservation pattern of microRNAs in cnidarians.

MINOR COMMENTS

- Reference formatting is inconsistent (sometimes numbering, sometimes Authors' name) and at least two references are missing for the list. Importantly, Moran et al 2017's review is not in the Reference list!

- The description of conserved target (lines 183-186) is not entirely clear to me. The figure associated is not much clearer in this context.

Referee: 2
Comments to the Author(s)
See attached.

Author's Response to Decision Letter for (RSPB-2019-2564.R0)

See Appendix B.

RSPB-2020-3169.R0

Review form: Reviewer 1

Recommendation

Accept as is

Scientific importance: Is the manuscript an original and important contribution to its field?

Good

General interest: Is the paper of sufficient general interest?

Good

Quality of the paper: Is the overall quality of the paper suitable?

Good

Is the length of the paper justified?

Yes

Should the paper be seen by a specialist statistical reviewer?

No

Do you have any concerns about statistical analyses in this paper? If so, please specify them explicitly in your report.

No

It is a condition of publication that authors make their supporting data, code and materials available - either as supplementary material or hosted in an external repository. Please rate, if applicable, the supporting data on the following criteria.

Is it accessible?

Yes

Is it clear?

Yes

Is it adequate?

Yes

Do you have any ethical concerns with this paper?

No

Comments to the Author

After re-reading the paper and the letter of response I can confirm that I am happy with the current version. The authors took into account and addressed my comment and/or justified their decision to keep the original analysis. As I wrote in my original report, despite the concerns I had, it is a valuable paper and meat for thought.

A last comment (yet not a request) to the authors is to ask whether they have checked for tandem duplications as a source of cnidarian microRNAs, as I understand this is a mechanisms that happens in plants but not in animals.

Decision letter (RSPB-2020-3169.R0)

18-Jan-2021

Dear Dr Moran

I am pleased to inform you that your manuscript RSPB-2020-3169 entitled "Conservation and turnover of miRNAs and their highly complementary targets in early branching animals" has been accepted for publication in Proceedings B.

The referee has recommended publication, but also suggest potential minor revisions to your manuscript. Therefore, I invite you to respond to the referee's comment and revise your manuscript. Because the schedule for publication is very tight, it is a condition of publication that you submit the revised version of your manuscript within 7 days. If you do not think you will be able to meet this date please let us know.

Sincerely,

Professor Hans Heesterbeek

Associate Editor

Board Member

Comments to Author:

I would like to thank the authors for addressing the issues raised in the previous submission. I agree with one of the referees on the fact that the current version has improved much. Congratulations on this great piece of work.

Reviewer(s)' Comments to Author:

Referee: 1

Comments to the Author(s).

After re-reading the paper and the letter of response I can confirm that I am happy with the current version. The authors took into account and addressed my comment and/or justified their decision to keep the original analysis. As I wrote in my original report, despite the concerns I had, it is a valuable paper and meat for thought.

A last comment (yet not a request) to the authors is to ask whether they have checked for tandem duplications as a source of cnidarian microRNAs, as I understand this is a mechanisms that happens in plants but not in animals.

Author's Response to Decision Letter for (RSPB-2020-3169.R0)

See Appendix C.

Decision letter (RSPB-2020-3169.R1)

25-Jan-2021

Dear Dr Moran

I am pleased to inform you that your manuscript entitled "Conservation and turnover of miRNAs and their highly complementary targets in early branching animals" has been accepted for publication in Proceedings B.

Open Access

Paper charges

Sincerely,
Proceedings B
<mailto:proceedingsb@royalsociety.org>

Appendix A

Praher et al. describe the microRNA (miRNA) complements of several cnidarian species, and focus on the important question surrounding the mechanism of targeting in cnidarians versus plants and bilaterian animals. They come to the important conclusion that cnidarian targeting is similar to plants, and the evolutionary dynamics are “intermediate” between plants and animals. Although this paper will be suitable for publication, there are a few things the authors need to address (and others they may wish to address) before then.

First and most importantly the reader does not seem to have access to the read and structural outputs for the accepted miRNAs. Given the sheer amount of garbage that has been published in the miRNA field over the last 15 years or so, this is an essential component to any paper such as this. Simply providing the mirdeep outputs for the accepted miRNAs would probably be good enough (making sure NOT to include mismatches in the outputs). Further, this would provide the necessary read data to evaluate arm switching (pg. 5).

Second, it is not clear (at least to me) how the miRNAs were annotated in all species - for example if a miRNA is “missing” in a taxon does that mean that a human has actually searched through both a genome and a small RNA library (if available for either)? Indeed, revising Fig. 1D would be good to indicate above each of the species if either a genome (G) or a transcriptome (T) is available for each indicated species. This is especially important as NEB-kits are very heterogeneous in microRNA yield, it would be useful how you accounted for potential false negatives.

Third, Fig. 1 needs to be redesigned to not only help those of us with failing eyes, but to better indicate the underlying evolutionary dynamics of cnidarian miRNAs.

Panel A. A comparison to our (Fromm et al. 2015) sequence logo for bilaterians would be really useful here I think. For example, we too (of course) highlight the U bias in position 1, but there is also a strong bias for U in position 9, something that is not seen here. Using a Chi-square test would also indicate significant deviations (if any). Also, the panel is much too small!

Panel B. Again, much too small! Also, I think I sent you the following, which might be worth considering when comparing the sequence evolutionary dynamics between cnidarians and bilaterians:

Slicing vs. Seed-based Targeting in Metazoans

Of course, my numbers are out of date, but the difference is striking and a reanalysis of this sort on your end would really strengthen your arguments.

Panel C. This should have an evolutionary tree on the top connecting each of the species (as in D), and to organize the miRNAs as apomorphically as opposed to numerically so it can be easier seen how the evolution of gains is working in this group.

Panel D. The key needs to be moved away from C, as it looks like it goes with C and not D. Indicate G vs T at the top, and also indicate the losses. And a couple of minor points: you use a butterfly as the bilaterian vignette but *Dre* for a sequence - use the same bilaterian (*D. melanogaster* or human, for example). Also, please note that there are 31 bilaterian families (<https://mirgenedb.org/family?node=4>) not 35.

Fourth, I'm very confused about whether degradome data were acquired for other taxa beyond *Nematostella* - it seems based on lines 200-203 that they are, but nothing is figured. Please clarify.

Fifth, a mistake made on pg 9 is that the *Mir-96/HoxB* relationship is not the oldest known targeting relationship, the *Let-7/Lin-41* is as it's found in both nematodes and vertebrates (and likely elsewhere including fly, but I don't know this for sure).

Sixth, Figure 2 could also use a slight re-design.

Panel A. Again, a phylogenetic tree at the top linking the species would be helpful. Also the spacing of the letters of the targets is a bit messed up.

Panel C. Please change the A+ and C* to something else (like just indicate both of the nucleotides on the sequence). I originally read this as "amino" and "carboxy" so needless to say I was a bit confused! :) Also, please label the ends of your sequences so the reader knows the orientation.

Minor points:

- 1) On page 5 (line 126) you write, “each family consisting of several members.” Can you clarify - the bilaterian LCA had single members for most of these families, but virtually all have more than one given (in part) the genome duplications in vertebrates.
- 2) Line 195 you write, “in contrast to Bilateria.” But as you point out Mir-196, as well as Mir-675 are known to work via slicing, so please rewrite slightly to indicate that although rare, it’s not about cnidarians vs. bilaterians but instead how each taxon utilizes a specific targeting strategy.
- 3) You could just reference Erwin et al. (2011) for most of your molecular clock estimates as we looked comprehensively across Metazoa rather than separate studies for each metazoan clade.
- 4) There is a typo in ref. 9 (the question marks).

Appendix B

Dear Editor,

We hereby submit a revision for our manuscript titled “Conservation and turnover of miRNAs and their highly complementary targets in early branching animals” (RSPB-2019-2564). We have substantially edited the manuscript following the remarks of the reviewers and expanded our analysis to additional cnidarian species. One major change we introduced in the focus of an experimentally-supported set of bona fide miRNAs from *Nematostella vectensis* that was attained by immunoprecipitation of Argonaute proteins (this was recently published, Fridrich et al. 2020 Nature Communications 11: 6187). This list enabled us to make a much more accurate comparative analysis in the other species in our current manuscript and we believe our results are much improved now. We also decided to drop any direct comparisons of the evolutionary pace of death-and-birth of miRNAs in bilaterians, cnidarians and plants as we realized that differences between different studies in prediction methods and annotation criteria can affect those numbers quite significantly.

Please find below our detailed responses to the comments by the referees which we provide in a point-by-point style. We are thankful for the detailed and helpful comments provided by both reviewers and we hope you will find our revision satisfactory.

Many thanks and best regards,

Daniela Praher, Yehu Moran and Ulrich Technau

Reviewer 1:

Praher and collaborators characterized the evolutionary history of microRNAs in cnidarians and concluded that the turnover of microRNAs and their targets resembles more to that of plant microRNAs than to animals. This is explained in the context that plant and animal microRNAs may have a common evolutionary origin and cnidarians could show features of both. They also sequenced, discovered and annotated new microRNAs in cnidarians, which is always a valuable contribution to the microRNA community.

Although I still believe that the evidence supports a distinct evolutionary origin of microRNAs in plants and animals, I also believe that works like this, challenging the current paradigm, are important and must be always considered for publication. After all, the authors are known and respected and their view and analyses are always welcomed. However, in this occasion I think there is a major issue in the analysis that must be addressed (see below). For that reason, I cannot recommend its publication and I suggest to the editor that he rejects the publication of the paper in its current form. Nevertheless, if the authors consider my concerns and the results still hold, I'll be happy to evaluate this paper again.

MAJOR COMMENTS

The main conclusion that microRNA turnover in cnidarians is similar to that in plants (that is, very high) compared to animals, is based on a strict criteria to determine microRNA orthology. The authors consider only those microRNAs annotated in the miRDeep2 pipeline which can be very strict. On the contrary, other animal microRNAs have been annotated using more ‘relaxed’ criteria (that is, BLAST) followed from extensive manual curation. That may result in a different annotation depth between this and other studies. Specifically, these are the things I think the authors should consider:

- The coverage of the genome sequence of cnidarians is lower than that of other animals. The authors acknowledge that but, how would this affect to any annotation/comparison? A more in depth comparative analysis, accounting for differences in sequencing coverage, is needed.

Response: Actually, for *Nematostella vectensis*, *Exaiptasia pallida*, *Acropora digitifera* and *Acropora millepora* the sequencing coverage is not low at all. Most of those genomes were sequenced by the classic Sanger method and achieved pretty good integrity as indicated by the fact that the vast majority of bilaterian genes had orthologs present in those genomes and the N50 values were decent (for example 470 kbp for *Nematostella* and 191 kbp for *A. digitifera*, please see Putnam et al. 2007 Science and Shinzato et al. 2011 Nature). These scaffold lengths are more than enough for annotating miRNA precursors that tend to be less than 100 bp long. Thus, we do not believe that the length of the scaffold comprises any substantial problem in our analysis. The only exception is *Anemonia viridis*, as we assembled its genome based on short reads, but this is only one species and we notify the reader regarding this fact.

- The authors claim (here and elsewhere) that cnidarian microRNAs show features of animal and plant microRNAs. However, the annotation criteria used is that of Fromm et al, which is specifically designed for vertebrates (as a matter of fact, it was originally devised for human microRNAs). How would this affect to the annotation of microRNAs in Cnidarians?

Response: While some features of cnidarian miRNAs indeed seem to be plant-like (e.g., their mode of action), others are very much “animal-like”. Cnidarians carry the microprocessor components Droscha and Pasha and the cropping step in miRNA biogenesis seems to be very similar to that of bilaterians. Cnidarians do have endogenous siRNAs, and it seems that their miRNAs can evolve from siRNA-like precursors, but those are loaded into another Argonaute protein and do not exhibit evolutionary conservation between different anthozoan species (please see Fridrich et al. 2020 Nature Communications for more details).

- The authors evaluate cnidarian microRNA targets by long pairwise complementarity, like in plants, but they also acknowledge that short targets (like in animals) can have also an impact. Wouldn't this contradict the main message of the paper?

Response: We actually do not acknowledge that. We now know for a fact that “animal like” miRNA-target matches that are restricted to the seed sequence (positions 2-8 of the miRNA) are non-functional in *Nematostella*. We have assayed this experimentally and we confidentially share this unpublished data with the reviewers and the editor. In the following graph you can see that an mRNA encoding mCherry and carrying three seed-restricted matches to a co-injected mimic miRNA (“mimiR”) do not cause any measurable decrease in expression in *Nematostella* (measured in ELISA

with anti-mCherry) whereas a single “plant-like” site for the same miRNA (“positive control”) results in a very strong downregulation of the mCherry expression.

We also have additional experimental data showing that short matches that lack pairing at the middle of the miRNA do not exert any effect in cnidarians. This is also supported by the comparison presented in the new Figure 2C where it is noticeable that whereas in bilaterians substitutions at positions at the middle of the miRNA are common, they are very rare in cnidarians.

Last, but not least, I selected one example of a microRNA that is conserved in some cnidarians but not in others: mir-2050. This microRNA is not detected in *Stylophora pistillata* with the author’s methodology. However, a genomic blast reveals that a putative ortholog is detectable in the genome.

Response: The Reviewer is absolutely correct, miR-2050 exists in the genome of *Stylophora* and is expressed in this species. We are thankful to the Reviewer for noticing this error as it made us notice a problem in our workflow that we now fixed.

MINOR COMMENTS

- Reference formatting is inconsistent (sometimes numbering, sometimes Authors’ name) and at least two references are missing for the list. Importantly, Moran et al 2017’s review is not in the Reference list!

Response: Thank you for noticing this. We have fixed these errors.

- The description of conserved target (lines 183-186) is not entirely clear to me. The figure associated is not much clearer in this context.

Response: This was thoroughly revised and we hope it is now much clearer.

Reviewer 2

Praher et al. describe the microRNA (miRNA) complements of several cnidarian species, and focus on the important question surrounding the mechanism of targeting in cnidarians versus plants and bilaterian animals. They come to the important conclusion that cnidarian targeting is similar to plants, and the evolutionary dynamics are “intermediate” between plants and animals. Although this paper will be suitable for publication, there are a few things the authors need to address (and others they may wish to address) before then.

First and most importantly the reader does not seem to have access to the read and structural outputs for the accepted miRNAs. Given the sheer amount of garbage that has been published in the miRNA field over the last 15 years or so, this is an essential component to any paper such as this. Simply providing the mirdeep outputs for the accepted miRNAs would probably be good enough (making sure NOT to include mismatches in the outputs). Further, this would provide the necessary read data to evaluate arm switching (pg. 5).

Response: We agree with the Reviewer and provide now the requested information in Supplementary Files S6-S10.

Second, it is not clear (at least to me) how the miRNAs were annotated in all species - for example if a miRNA is “missing” in a taxon does that mean that a human has actually searched through both a genome and a small RNA library (if available for either)? Indeed, revising Fig. 1D would be good to indicate above each of the species if either a genome (G) or a transcriptome (T) is available for each indicated species. This is especially important as NEB-kits are very heterogeneous in microRNA yield, it would be useful how you accounted for potential false negatives.

Response: We apologize, if the methodology of miRNA annotation in our study was not clear. We changed the wording in figure 1 and the categories now are “miRNA expressed” and “miRNA not expressed”. If a miRNA is “not expressed”, it is a) not detected by the core algorithm and the quantifier module of miRDeep2 for species with available genome or b) not detected by the quantifier module alone for species, where only a transcriptome is available, but not a genome. The only exception is miRNA-2036 in *Acropora digitifera*: For this miRNA reads are detected but it was excluded since it did not pass our manual inspection criteria adapted from Fromm et al., 2015 (it had a 5' homogeneity for the mature miRNA strand of only 58%). A thorough redesign of this figure was accomplished and it is now indicated for which species a genome or a transcriptome is available as suggested by the reviewer.

Third, Fig. 1 needs to be redesigned to not only help those of us with failing eyes, but to better indicate the underlying evolutionary dynamics of cnidarian miRNAs.

Panel A. A comparison to our (Fromm et al. 2015) sequence logo for bilaterians would be really useful here I think. For example, we too (of course) highlight the U bias in position 1, but there is also a strong bias for U in position 9, something that is not seen here. Using a Chi-square test would also indicate significant deviations (if any). Also, the panel is

much too small!

Response: We have increased the size of the panel and redesigned the figures following the Reviewer's advice. We refrained from a statistical comparison for the logo representations as basically nothing besides position 1 shows any meaningful enrichment in cnidarians.

Panel B. Again, much too small! Also, I think I sent you the following, which might be worth considering when comparing the sequence evolutionary dynamics between cnidarians and bilaterians:

Response: We highly appreciate the Reviewer's advice and generated a figure similar to the one he suggested (Now Figure 2C). However, in order to have fair comparisons, we took the same number of anthozoan and bilaterian species (nine, maximally available for anthozoans) and the same number of conserved miRNAs (nine, again the maximum available for anthozoans) for the comparison because taking the maximum available data will generate a very serious sampling bias as there is much more data for bilaterians than cnidarians which can inflate the numbers in a disproportional way. This analysis shows that cnidarian miRNAs tend to have hardly any mutations in the central positions (7-18), while bilaterian miRNAs have very few mutations in the seed region (2-8), as shown in many previous analyses. This independently supports our previous claim that cnidarian miRNAs bind with (almost) full complementarity and there is no specific selection pressure on the seed region compared to more central regions.

Fourth, I'm very confused about whether degradome data were acquired for other taxa beyond Nematostella - it seems based on lines 200-203 that they are, but nothing is figured. Please clarify.

Response: we apologize for this confusion. There is no degradome data available for other cnidarians and we hope that the new version of the manuscript is clearer at this point (please see line 181 in the text).

Fifth, a mistake made on pg 9 is that the Mir-96/HoxB relationship is not the oldest known targeting relationship, the Let-7/Lin-41 is as it's found in both nematodes and vertebrates (and likely elsewhere including fly, but I don't know this for sure).

Response: We would argue that the Let-7/Lin-41 case is less clear due to the lower conservation level, but to mitigate this issue in the revised text we do not claim that miR96/HoxB is the oldest one, but just one of the oldest known cases.

Sixth, Figure 2 could also use a slight re-design.

Response: we substantially revised the figures following the comments of the Reviewer.

Minor points:

1) On page 5 (line 126) you write, "each family consisting of several members." Can you clarify - the bilaterian LCA had single members for most of these families, but virtually all have more than one given (in part) the genome duplications in vertebrates.

Response: this part of the text was substantially edited.

2) Line 195 you write, "in contrast to Bilateria." But as you point out Mir-196, as well as Mir-

675 are known to work via slicing, so please rewrite slightly to indicate that although rare, it's not about cnidarians vs. bilaterians but instead how each taxon utilizes a specific targeting strategy.

Response: this part of the text was edited.

3) You could just reference Erwin et al. (2011) for most of your molecular clock estimates as we looked comprehensively across Metazoa rather than separate studies for each metazoan clade.

Response: Thank you for the suggestion. We refer both to Erwin et al. and to other studies as well.

4) There is a typo in ref. 9 (the question marks).

Response: Thank you for noticing this typo.

Appendix C

Dear Editor,

We were very pleased to learn that you and the reviewers found our revision improved and that we successfully addressed all the previous comments. We would like to take this opportunity to thank again you and the two anonymous reviewers for your help in improving this manuscript.

We address below the last remaining remark by Reviewer 1.

Many thanks and best regards,

Daniela Praher, Yehu Moran and Ulrich Technau

Comment: After re-reading the paper and the letter of response I can confirm that I am happy with the current version. The authors took into account and addressed my comment and/or justified their decision to keep the original analysis. As I wrote in my original report, despite the concerns I had, it is a valuable paper and meat for thought.

A last comment (yet not a request) to the authors is to ask whether they have checked for tandem duplications as a source of cnidarian microRNAs, as I understand this is a mechanisms that happens in plants but not in animals.

Response: We are not entirely sure what the Reviewer means by tandem duplications in this specific case. If they mean that tandem duplication of miRNA precursor expands the miRNA repertoire, this is in fact something uncommon in plants but very common in bilaterians. In Cnidaria this is a very minor phenomenon as the vast majority of miRNAs have a single copy in cnidarian genomes.

However, some exceptions do occur such as miR-2024 in *Nematostella vectensis* and miR-2022 in *Exaiptasia pallida*. The case of miR-2024 in *Nematostella vectensis* was described in a previous publication of ours (Moran et al., Genome Res. 2014).

If on the other hand the Reviewer means that miRNAs can be born by inverted duplications of their own target genes, this is indeed a significant phenomenon happening both in plants and cnidarians but not in bilaterians. We already addressed this in detail in a recent paper (Fridrich et al. 2020 Nature Communications 11: 6187).

In any case, despite the interest in this point we feel that it is somewhat off-topic. Furthermore, as we are already currently at the strict length limits of *Proceedings B* (12 printed pages), including this in the text would necessarily mean removing something else. Thus, we would prefer not to include it in our current manuscript and we hope for your understanding.